# Distinct Clusters of Testosterone Levels, Symptoms, and Serum Trace Elements in Young Men: A Cross-Sectional Analysis

**DOI:** 10.3390/nu17050867

**Published:** 2025-02-28

**Authors:** Takazo Tanaka, Kosuke Kojo, Takahiro Suetomi, Yoshiyuki Nagumo, Haruhiko Midorikawa, Takaaki Matsuda, Ayumi Nakazono, Takuya Shimizu, Shunsuke Fujimoto, Atsushi Ikeda, Shuya Kandori, Hiromitsu Negoro, Tatsuya Takayama, Hiroyuki Nishiyama

**Affiliations:** 1Department of Urology, Institute of Medicine, University of Tsukuba, 1-1-1 Tennodai, Tsukuba 305-8575, Japan; 2Center for Human Reproduction, International University of Health and Welfare Hospital, 537-3, Iguchi, Nasushiobara 329-2763, Japantakayama@iuhw.ac.jp (T.T.); 3Tsukuba Clinical Research & Development Organization, University of Tsukuba, 2-1-1, Amakubo, Tsukuba 305-8576, Japan; 4Department of Urology, Ibaraki Seinan Medical Center Hospital, 2190, Sakai-machi, Sashima-gun, Ibaraki 306-0433, Japan; 5Department of Psychiatry, University of Tsukuba Hospital, 2-1-1, Amakubo, Tsukuba 305-8576, Japan; 6Department of Endocrinology and Metabolism, University of Tsukuba Hospital, 2-1-1 Amakubo, Tsukuba 305-8576, Japan; 7Health Care Analysis Center, Renatech Co., Ltd., 4-19-15, Takamori, Isehara 259-1114, Japan; 8Department of Urology, International University of Health and Welfare Hospital, 537-3, Iguchi, Nasushiobara 329-2763, Japan

**Keywords:** androgen deficiency in young men, biomarkers of stress, EHS, Heinemann’s AMS rating scale, inductively coupled plasma mass spectrometry, libido and erectile dysfunction, male hypogonadism, semen analysis, reproductive endocrinology, Zung’s Self-Rating Depression Scale

## Abstract

**Background/Objectives**: Modern societal stressors have been linked to declining testosterone levels among young men, contributing to somatic, psychological, and sexual health problems. Despite growing evidence suggesting a link between trace elements and testosterone-related symptoms, there are only a few comprehensive analyses on younger populations. This study’s aim was to examine how serum trace elements modulate the relationship between testosterone levels and symptom severity. **Methods**: This cross-sectional study included 225 young men seeking infertility consultation in Japan. Serum total and free testosterone levels were measured, along with self-reported symptoms using the Aging Males’ Symptoms scale (somatic, psychological, sexual) and the Erection Hardness Score. The serum concentrations of 20 trace elements were measured. We used unsupervised clustering to classify participants based on testosterone levels and symptom severity and then compared the distribution of trace elements among the resulting clusters. **Results**: Three distinct clusters emerged: (1) lowest testosterone with highest symptom severity, (2) intermediate, and (3) highest testosterone with minimal symptoms. Interestingly, the intermediate cluster displayed low testosterone levels but minimal symptoms. Eleven trace elements (phosphorus, sulfur, potassium, calcium, iron, zinc, arsenic, rubidium, strontium, molybdenum, and cesium) were identified as potential contributors to testosterone dynamics. Weighted quantile sum regression indicated that phosphorus, strontium, and molybdenum negatively influenced testosterone outcomes, whereas iron, sulfur, and zinc were beneficial. **Conclusions**: Serum trace element profiles are significantly associated with testosterone levels and symptom severity in young men. Targeted interventions may address testosterone decline and its implications. These findings may help develop tailored strategies for optimizing male health.

## 1. Introduction

Since the mid-20th century, global industrialization has successfully mitigated many infectious diseases and poverty-related issues [1]; however, it has introduced a new set of stressors that adversely affect men’s physical and mental health. Despite advances in healthcare, men continue to lag behind women in life expectancy, underscoring the significance of sex-specific approaches to disease prevention and management [2]. Unhealthy lifestyle patterns—such as poor dietary habits [3,4], sedentary lifestyle [5], and socioeconomic disparities [6,7,8]—along with traditional masculinity norms [9,10,11], have collectively contributed to higher risks of chronic fatigue [12], depression [13], and erectile dysfunction [14,15].

Similarly, serum testosterone levels have shown a significant decline over the years, even in younger men, with one report noting a steady drop among males aged 15–39 in the United States from 1999 to 2016 [16]. Stress not only directly affects health [17] but also influences the hypothalamic–pituitary–gonadal axis, a critical component of reproductive endocrinology, leading to reduced testosterone secretion [18]. This reduction is associated with symptoms such as obesity, decreased muscle strength, reduced bone density, depressive symptoms, cognitive decline, and impaired sexual function, potentially leading to cardiovascular diseases and increased risk of mortality [19]. Research on severe testosterone deficiency (hypogonadism) has mainly focused on older men [20] or on pediatric and young adult populations with clearly defined endocrine disorders [21,22]. Only a few studies have examined how milder testosterone declines influence somatic, psychological, and sexual well-being in apparently healthy young men [23,24,25]. Despite an apparent increase in the number of young individuals experiencing varying degrees of somatic, psychological, and sexual health concerns without seeking medical attention, there remains a clear lack of studies examining the relationship between testosterone levels and the severity of these health issues. Moreover, large population-based surveys indicate that a considerable portion of men with low testosterone levels remain asymptomatic, whereas others with ostensibly “normal” levels exhibit pronounced symptoms of fatigue, depressed mood, and diminished libido [26].

Emerging evidence suggests that trace elements play a role in modulating testosterone production and related symptoms. For example, deficiencies or surplus of essential neurometals (e.g., iron (Fe), zinc (Zn), copper (Cu), and manganese (Mn)) [27] can aggravate fatigue [28], mood disorders [29], and even contribute to diminished testicular function [30]. However, most research to date has focused on single-element effects in older or clinically hypogonadal populations, leaving a knowledge gap regarding the influence of multiple trace elements on testosterone levels and symptom severity in younger men. This gap is notable given the mounting reports of youth experiencing stress-related health issues and potential mild testosterone decline without overt hypogonadism.

Therefore, this study aimed to investigate whether distinct testosterone levels and clinical symptoms are associated with different serum trace element concentrations in a cohort of predominantly young men. We enrolled participants attending an infertility clinic, ensuring a broad range of testosterone values and symptom severities, including those with normal testicular function. By harnessing unsupervised clustering [31] to classify participants based on testosterone levels and symptom severity, rather than relying on rigid cut-off definitions, we sought to capture a continuous spectrum of hormonal and symptomatic variability. We then compared the serum trace element levels among the resulting clusters to identify potential elemental influences on male reproductive and overall health. This approach offers new insights into how emerging stressors and environmental exposures may contribute to men’s health challenges in modern society.

## 2. Materials and Methods

### 2.1. Study Design

#### 2.1.1. Cross-Sectional Study

This exploratory cross-sectional study retrospectively analyzed data from a project designed to develop new diagnostic strategies for male infertility [32]. Participants were recruited between August 2019 and April 2022 at a single specialized male infertility outpatient clinic at the International University of Health and Welfare Hospital (Nasushiobara, Japan).

#### 2.1.2. Inclusion and Exclusion Criteria

The inclusion criteria were (1) availability of serum total and free testosterone measurements and (2) completed self-reported questionnaires (Aging Male’s Symptoms (AMS) scale and Erection Hardness Score (EHS)) assessing symptoms related to lower testosterone levels and erectile function. Participants were excluded if serum trace element data were unavailable or if seminal trace element data were missing (e.g., due to complete semen utilization for sperm retrieval in cases of azoospermia). Patients with a history of malignancy [33] were also excluded. After applying these criteria, 225 participants were selected for analysis (Appendix A).

#### 2.1.3. Outcomes

The primary outcome was to classify participants into clusters based on serum total and free testosterone levels and symptom severity and to subsequently compare serum trace element concentrations among these clusters. The secondary outcomes included age, anthropometric measurements, testicular volume, semen parameters, and reproductive hormones (luteinizing hormone, follicle-stimulating hormone, prolactin, and estradiol levels). Given that serum trace element concentrations vary with age [34,35] and that our focus was on younger men, we performed propensity score matching for age, resulting in a final cohort of 168 men retained for the analysis of serum trace elements (Appendix A).

### 2.2. Assessment of Serum Testosterone and Testosterone-Related Hormone Levels

Serum total testosterone level was measured using a chemiluminescent enzyme immunoassay (Elecsys Testosterone II, Roche Diagnostics K.K., Tokyo, Japan), and free testosterone level was measured using a radioimmunoassay (Free Testosterone RIA kit “SML”, DENIS Pharma K.K., Tokyo, Japan). Both assessments were outsourced to a certified contract laboratory (SRL Inc., Tokyo, Japan) [36].

In healthy young men without hypogonadism, testosterone levels typically peak in the early morning and decrease by approximately 10% in the evening [37]. However, because the primary aim of this cross-sectional study was to detect abnormally low testosterone levels, blood collection times were not strictly standardized. Nonetheless, with a few exceptions, samples were drawn between 8:00 AM and 2:00 PM; therefore, any diurnal variation was deemed acceptable for our purposes.

Free testosterone is a sensitive indicator of symptom severity in Japanese men [38], as free testosterone correlates closely with symptom severity [39]. However, it is crucial to note that these values are significantly lower, approximately one-seventh, than the theoretically calculated free testosterone values commonly used in international practice [40].

Luteinizing hormone, follicle-stimulating hormone, prolactin, and estradiol levels were measured using chemiluminescent immunoassays: Chemilumi LH (Siemens Healthcare Diagnostics, Tarrytown, NY, USA), ARCHITECT FSH (Abbott Laboratories, Abbott Park, IL, USA), ARCHITECT Prolactin (Abbott Laboratories, Abbott Park, IL, USA), and ARCHITECT Estradiol II (Abbott Laboratories, Abbott Park, IL, USA), respectively.

### 2.3. Analytical Approach to Testosterone Levels

Although cut-off values for testosterone deficiency have been proposed [41,42,43], these thresholds may not reflect continuous variations. Therefore, unsupervised clustering [31] was employed to classify participants without relying on pre-defined cut-off values.

### 2.4. Symptom Evaluation

Participants completed the Japanese version of Heinemann’s AMS scale to evaluate somatic, psychological, and sexual symptoms [44,45]. Erectile function was assessed using the EHS [46,47], and depressive symptoms were evaluated using Zung’s Self-Rating Depression Scale (SDS) [48,49].

### 2.5. Measurement of Serum Trace Element Concentrations

This study collected an additional 6 mL of blood during serum hormone screening to create a comprehensive dataset of serum trace element concentrations. We employed inductively coupled plasma mass spectrometry (Agilent 7800, Agilent Technologies, Santa Clara, CA, USA) to quantify trace element levels, following our previously published protocols [50,51,52]. Briefly, after the serum had coagulated, it was separated by centrifugation to remove blood cells and then digested with nitric acid and hydrogen peroxide under heat. The concentrations of 20 trace elements were then measured: lithium (Li), sodium (Na), magnesium (Mg), phosphorus (P), sulfur (S), potassium (K), calcium (Ca), Mn, Fe, cobalt (Co), Cu, Zn, arsenic (As), selenium (Se), rubidium (Rb), strontium (Sr), molybdenum (Mo), cesium (Cs), barium (Ba), and thallium (Tl).

### 2.6. Semen Analysis and Other Secondary Outcome Measures

Height, weight, and body mass index (BMI) were recorded. Testicular volume was measured using a punched-out orchidometer [53]. Semen parameters (volume, sperm concentration, and motility) were obtained following the World Health Organization (WHO) 2010 guidelines [54]. Although the WHO guidelines were updated in 2021, these revisions did not affect the analytical methods used in this study. Trace elements derived from seminal plasma have been analyzed previously [32] but were beyond the scope of this investigation.

### 2.7. Statistical Analysis

#### 2.7.1. Preparation of Variables

Six explanatory variables (total testosterone, free testosterone, AMS somatic, AMS psychological, AMS sexual, and EHS) were selected for dimensionality reduction and clustering and were standardized to Z-scores. Serum trace element concentrations used in multivariate regression analyses were applied to Box–Cox transformation [55].

#### 2.7.2. Dimensionality Reduction and Clustering

Principal component analysis (PCA) [56] was performed for dimensionality reduction, followed by k-means clustering [31] to identify subgroups without relying on pre-defined cut-off values. Standard approaches, including the elbow method and silhouette analysis, were employed to determine the appropriate number of clusters [57]. Although no single optimal value of k emerged, we selected k = 3 as an exploratory choice to balance statistical considerations with subsequent clinical interpretability [58]. Based on the prior literature [59], we anticipated that the resulting clusters would span a range of testosterone levels and symptom severity, potentially including a group with high testosterone and minimal symptoms, one with low testosterone and severe symptoms, and an intermediate cluster with characteristics falling between these two extremes.

#### 2.7.3. Age Adjustment and Comparison of Serum Trace Element Concentrations

Age differences among the clusters were adjusted using propensity score nearest-neighbor matching to minimize age-related effects as confounding factors influencing trace element homeostasis [60]. Spearman’s rank correlation coefficients were calculated between the explanatory variables and each of the 20 trace elements. Trace elements meeting a pre-defined correlation threshold (≥0.1 [61] and a significant *p*-value) or showing significant differences among clusters based on Kruskal–Wallis tests were selected for further multivariate analyses.

#### 2.7.4. Multivariate Regression Analyses

Ordinal logistic regression analysis [62] was used to evaluate the associations between trace elements and cluster membership. Clusters were treated as ordinal variables, reflecting a spectrum from healthier (higher testosterone and fewer symptoms) to less healthy (lower testosterone and more severe symptoms).

Given the highly intercorrelated nature of trace element data, we further applied weighted quantile sum (WQS) regression [63] to identify the most influential trace elements. WQS regression is particularly suited for modeling complex mixtures and identifying key contributors [64]. Because WQS regression compares two groups at a time, we performed four distinct comparisons to validate the consistency of trace element contributions across the ordinal progression of testosterone levels and symptoms.

#### 2.7.5. Software and Statistical Significance

All statistical analyses and graphical representations were performed using R software (version 4.2.2; R Foundation for Statistical Computing, Vienna, Austria). Ordinal logistic regression was conducted using the MASS package (version 7.3–58.1), and WQS regression was performed using the gWQS package (version 3.0.5). Statistical significance was set at a *p*-value of <0.05 (two-sided).

For non-normally distributed data, Spearman’s rank correlation coefficient (Spearman’s rho) and Kruskal–Wallis tests were employed for correlations and comparisons, respectively, across the three clusters. The Wilcoxon rank-sum test with Bonferroni correction was used for pairwise comparisons. Because the EHS comprises only four discrete values, medians and interquartile ranges are not always intuitive. Therefore, for ease of interpretation, EHS results are presented as means with standard errors [65].

## 3. Results

### 3.1. Patient Characteristics and Correlations Among Explanatory Variables

A total of 225 participants were included in the study. Table 1 summarizes the baseline characteristics of the study population. The median age was 37 years (range: 22–75 years), with most participants falling in the adolescent and young adult age groups. Approximately 50% of the participants exhibited semen parameters, such as semen volume, sperm concentration, sperm motility, and total sperm count, below the WHO 2021 reference standards (Appendix A).

Of the 225 participants, 17% had serum testosterone levels below the lower limit of the reference intervals (cut-off values) typically applied in Japan for either total testosterone or free testosterone (Appendix A). Additionally, 58% of the participants reported AMS scores classified as mild (27 or higher) or more severe. The mean (standard error) for the EHS was 3.44 (0.05), with 6.2% of the participants reporting an EHS grade of ≤2 (Appendix A).

Significant correlations were observed between total and free testosterone levels (rho = 0.57). Among the AMS subdomains, only the sexual subdomain showed a significant negative correlation with both total and free testosterone levels (rho = −0.17 for each). Furthermore, all AMS subdomains were significantly intercorrelated, as were AMS scores and EHS (Appendix A).

### 3.2. Dimensionality Reduction, Clustering, and Baseline Characteristics of Explanatory Variables and SDS

To examine the overall importance of multiple explanatory variables in men seeking to have children, we first performed PCA on the standardized data. The first principal component (PC1) primarily captured the variance related to symptom severity. All AMS subdomain scores were positively weighted on PC1, whereas the EHS score was negatively weighted (Figure 1a). The second principal component (PC2) reflected male sex hormone-related variables, with serum total and free testosterone levels showing positive correlations (Figure 1a).

Following PCA, we employed k-means clustering to classify the participants. To determine the number of clusters (k), we initially explored k = 2, 3, 4, and 5 using standard methods. The elbow method did not reveal a clear inflection point (Appendix A), and silhouette analysis yielded moderate or low silhouette coefficients across all tested k-values (Appendix A). Despite the absence of a single definitive optimal k, we selected k = 3 as a balanced approach, allowing for a clinically meaningful interpretation of the results.

With k = 3, three clusters revealed inherent patterns in the data (Figure 1a). Cluster 1 consisted of participants with the lowest total and free testosterone levels and the highest symptom severity. Cluster 3 comprised participants with the highest testosterone levels and minimal symptoms, whereas Cluster 2 represented an intermediate group with characteristics bridging the gap between Clusters 1 and 3.

A univariate comparison of the cluster profiles is presented in Figure 1b. Total and free testosterone levels were significantly lower in Clusters 1 and 2 than in Cluster 3 (both *p* < 0.001), although there were no significant differences between Clusters 1 and 2 (both *p* > 0.05). AMS somatic, psychological, and sexual subdomain scores were significantly higher in Cluster 1 than in Clusters 2 and 3 (all *p* < 0.001). Notably, somatic and psychological scores did not differ significantly between Clusters 2 and 3 (all *p* > 0.05). Although Cluster 2 shared similarly low testosterone levels as Cluster 1, its somatic and psychological symptoms resembled those of Cluster 3 (all *p* > 0.05). The EHS values decreased in the order of Cluster 3, Cluster 2, and Cluster 1 (*p* = 0.038).

In summary, although the elbow and silhouette methods did not identify a single “optimal” k, the choice of k = 3 resulted in clusters with distinct clinical and hormonal profiles, providing a meaningful framework for subsequent analyses.

Before age adjustment, significant differences were observed among the clusters in terms of age, explanatory variables, SDS scores, and height (*p* = 0.010) (Table 2 and Table 3). To ensure that the differences in serum trace element concentrations were not driven by age disparities, propensity score nearest-neighbor matching was applied, resulting in an age-matched subset of 168 participants (Appendix A). After age adjustment, significant differences in the explanatory variables and SDS persisted, except for the EHS (Table 2), whereas no significant differences were identified in semen parameters or other secondary outcomes (Table 3). This comprehensive evaluation of explanatory variables and SDS (Table 2), along with semen parameters and other secondary measures (Table 3), established a nuanced baseline for understanding the interplay between testosterone levels, symptoms, and subsequent trace element analyses.

### 3.3. Identification of Trace Elements Associated with Cluster Classification

Following age adjustment, we detected significant differences in serum trace element concentrations among the clusters for S, K, Ca, As, Rb, and Cs (*p* = 0.029, 0.014, 0.009, 0.019, and 0.049, respectively) (Table 4). Spearman’s rank correlation coefficient between the 6 explanatory variables and the 20 measured trace elements in the age-adjusted cohort revealed that P, S, K, Ca, Fe, Zn, As, Rb, Sr, and Mo were significantly correlated (rho ≥ 0.1, *p* < 0.05) with at least one explanatory variable (Figure 2).

To identify which trace element independently influenced cluster membership, we applied ordinal logistic regression (Figure 3), treating the clusters as ordinal variables reflecting a sequential decline from the testosterone advantage (from Cluster 3 to Cluster 1). Of the 11 trace elements identified through univariate comparisons and correlation analysis (P, S, K, Ca, Fe, Zn, As, Rb, Sr, Mo, and Cs), Mo showed a significant association with lower cluster membership toward Cluster 1 (odds ratio (OR) = 1.42, 95% confidence interval (CI): 1.01–2.01, *p* = 0.045), indicating that higher Mo levels aligned with lower testosterone and more severe symptoms. Conversely, Fe was significantly associated with higher cluster membership in Cluster 3 (OR = 0.70, 95% CI: 0.51–0.96, *p* = 0.030), suggesting a positive role in maintaining higher testosterone levels and fewer symptoms. Although P and As showed trends, they did not reach statistical significance in ordinal logistic regression (OR = 1.25, 95% CI: 0.90–1.74, *p* = 0.182; OR = 0.72, 95% CI: 0.50–1.03, *p* = 0.071, respectively).

To further validate these findings, bidirectional WQS regression analysis was conducted (Figure 4). This analysis identified P, Sr, and Mo as consistent negative or neutral contributors to higher testosterone-related health (i.e., potentially associated with membership in lower testosterone clusters) and S, Fe, and Zn as consistent positive or neutral contributors (i.e., potentially supporting their beneficial roles in sustaining testosterone levels and mitigating symptoms). Notably, inconsistencies between the two modeling approaches were observed, indicating a more complex relationship between testosterone dynamics and symptom severity. Integrating both analyses underscored Mo as a key negative factor and Fe as a key positive factor, while P, Sr, S, and Zn also appeared to be influential in defining cluster characteristics.

## 4. Discussion

### 4.1. Overview of Clusters and Key Findings

This study provides insights into the interrelationship between testosterone levels, symptom severity, and serum trace element concentrations in young men. By applying unsupervised clustering [31], we identified three distinct groups that captured a continuum rather than relying on arbitrary hormonal cut-offs.

Cluster 1 showed low testosterone levels and severe symptoms, Cluster 2 showed low testosterone but minimal symptoms, and Cluster 3 showed higher testosterone with few complaints. The coexistence of low testosterone levels and minimal symptoms in Cluster 2 highlights that testosterone alone does not define a man’s health trajectory. Men differ in their susceptibility to low hormone levels, possibly due to differences in androgen receptor sensitivity, compensatory hormonal pathways, lifestyle factors, or psychological resilience [26]. Similarly, the high-symptom group may reflect complex interactions in which declining testosterone amplifies underlying stressors, such as fatigue, depression, or subclinical inflammation, rather than serving as the sole driver of poor health. These results reinforce the notion that a decline in testosterone levels alone does not uniformly translate into a symptomatic burden and highlight the complexity of individual susceptibility.

### 4.2. Trace Elements Associated with Testosterone Dynamics

Our findings demonstrate that trace elements may influence the variability in how testosterone levels, symptom severity, and serum trace element concentrations are interrelated among different individuals. After adjusting for age, we identified 11 trace elements (P, S, K, Ca, Fe, Zn, As, Rb, Sr, Mo, and Cs) as potential contributors to the dynamics of testosterone and health outcomes. Although elements such as P, Sr, and Mo were possibly linked to the likelihood of testosterone decline and more severe symptoms, Fe, S, and Zn appeared to be positively associated with healthier profiles.

#### 4.2.1. Mo, Sr, and Other Negative Contributors

Among the elements identified as negative contributors, Mo stood out for its negative contributions to testosterone-related outcomes, as indicated by both ordinal logistic and WQS regression. Multiple studies have documented negative correlations between Mo and serum testosterone [66], urinary testosterone [67], and the testosterone-to-LH ratio [68]. These conflicting findings emphasize the need to consider confounding factors [69,70,71,72]. Intriguingly, Mo may influence mood disorders, as it has been linked to both worsening [73,74] and alleviating [75,76] depressive symptoms.

Although Sr supports bone metabolism [77], its broader health implications remain unclear. Negative associations have been reported between urinary Sr and serum testosterone [78], as well as shifts in Sr levels during depression recovery [79] and in bipolar disorder [80]. In the present study, our WQS regression analysis indicated that Sr could negatively contribute to testosterone-related health when considered within a complex mixture of trace elements. This finding contrasts with the contradictory findings showing negative correlations of Sr with individual symptom severity indices (AMS somatic, psychological, and sexual) and its positive correlation with the EHS. This discrepancy suggests that the role of Sr may depend on interactions with other elements or non-linear dose–response relationships, warranting further investigation to clarify its overall impact.

P, S, Ca, and As also warrant further attention. P, S, As, and Se are classified as nonmetals; however, they are occasionally categorized as semiconductors or metalloids [81]. In contrast, Ca is classified as a metal.

For P, studies have revealed an inverse correlation with calculated free testosterone [82] and associations with increased cardiovascular risk [83] and erectile dysfunction [84]. P dysregulation has also been linked to stress and depressive symptoms [85]. While many studies have measured serum inorganic P concentrations, this study used inductively coupled plasma mass spectrometry to measure total P, encompassing both inorganic P and P-containing compounds [52]. Although a separate study has confirmed a correlation between inorganic P and total P concentrations [52], caution is warranted when interpreting the findings of this study.

S is critical in biological systems as part of S-containing amino acids [86]. In the present study, S correlated strongly with serum-free testosterone after adjusting for age and was associated with better health outcomes according to WQS regression. This link may partly stem from the influence of albumin on testosterone [87], as most S in serum is protein-bound [88,89], and albumin levels correlate with stress [90] and depressive symptoms [91]. Although Ca is a metal, it is often bound to albumin in the bloodstream, complicating interpretations [92]. In this study, As exhibited an unexpected behavior, showing a negative correlation with symptom severity, particularly with AMS somatic scores; this contrasts with its usual classification as a toxic element that impairs testosterone synthesis [93]. Animal studies suggest that As may have physiological roles via methionine metabolism [94]. These contradictions imply that confounding factors and context-specific mechanisms may shape the impact of As on male health.

#### 4.2.2. Fe, Zn, and Other Positive or Ambiguous Elements

Fe and Zn, prominent in andrology research [95,96], were the only elements besides S that consistently supported higher testosterone levels and better health outcomes in WQS analysis. Fe deficiency, a major cause of anemia, leads to fatigue and other symptoms [97] and harms testicular function [98]. Although testosterone may also modulate Fe homeostasis [99,100], Zn appears to buffer against Fe deficiency effects [101]. Early misdiagnosis of Zn deficiency as Fe-deficiency anemia [102] highlights the interplay between these two minerals. Even considering Zn’s correlation with albumin [103], numerous studies have confirmed a causal relationship between Zn and testosterone secretion [104]. Known as the “sex mineral” [105], Zn deficiency is linked to chronic pain [106], stress [107], depressive symptoms [108], and erectile dysfunction [109,110].

Prior reports have indicated positive effects of Mg [111], K [112,113], Co [114], and Se [115] on testosterone secretion, whereas Cu has been linked to negative effects [115]. Li [116,117], Mn [118], and Rb [119], as well as Cs, present more ambiguous associations, potentially driven by indirect relationships (e.g., correlations with serum K levels [120]). These elements may influence testosterone secretion in certain contexts, but their roles remain unclear within the framework of the current study.

Even elements traditionally considered essential for maintaining homeostasis may be associated with adverse outcomes, reflecting a delicate balance between their necessity and toxicity. Excessive or imbalanced levels can disrupt endocrine homeostasis [68,72]. Furthermore, the complex dose–response relationships that extend beyond this one-dimensional understanding cannot be ruled out; these elements may react differently in the presence of other nutritional and environmental factors. This complexity highlights the need to evaluate trace elements not in isolation but as essential components of a broader ecological system.

### 4.3. Clinical and Sociocultural Factors Affecting AMS, EHS, and Semen Parameters

#### 4.3.1. Semen Parameters and AMS

Despite the observed associations among testosterone levels, symptom severity, and trace elements, we found no significant differences in semen parameters between the identified clusters. This finding aligns with previous research [32], indicating that although testosterone is essential for spermatogenesis, its direct influence on semen quality may be limited [121]. Instead, trace elements may mediate or modulate the influence of testosterone on reproductive health, suggesting a more complex interplay than a simple hormonal effect on sperm parameters.

Although the AMS provides valuable insight into the overall symptomatic burden, differences in erectile function are generally mild. Research on AMS in the context of male infertility remains scarce, with only two Japanese studies examining this relationship [122,123]. Additionally, the presence of depressive symptoms, measured by the SDS, suggest that low testosterone levels may overlap with mood-related issues, highlighting the need to consider psychological factors when interpreting these findings.

#### 4.3.2. EHS Prevalence and Sociocultural Influences

Variations in EHS prevalence compared to other populations may stem from differences in study designs or participant characteristics. A more comprehensive understanding of these nuances likely requires the integration of sociocultural contexts and psychosocial variables that can influence both erectile and ejaculatory functions. For example, the rise of digital communication and social networking services has reshaped sexual behavior and interpersonal relationships, with recent nationwide surveys in Japan linking these societal changes to a high prevalence of reproductive challenges, such as reduced sexual interest and erectile dysfunction [124]. These shifts may amplify stress-related hormonal changes and alter the health trajectories of men.

In our study, the mean EHS exceeded 3.4 in all three clusters, and only 6.2% of participants had an EHS score of ≤2, paralleling the 6.1% prevalence of erectile dysfunction found in a 2015 national survey of men seeking infertility care [125]. However, these results differ from a 2024 national survey reporting over 10% of men aged 20–49 years with an EHS score of ≤2 [124]. One explanation for this discrepancy is that our participants, who explicitly sought clinical assistance to achieve paternity, likely had relatively stable relationships and favorable socioeconomic conditions. Conversely, the nationwide survey included a significant proportion of unmarried (39.1%) and unemployed (17.5%) men who may face different or more intense stressors. Indeed, married men generally exhibit lower salivary cortisol levels, a biomarker of stress, than unmarried men [126], and individuals who are unemployed or underemployed reportedly experience higher stress and depressive symptoms [127]. Such variations in participant demographics could greatly influence EHS findings.

#### 4.3.3. Symptom Disclosure and Ejaculatory Dysfunction

Another factor affecting AMS, EHS, and semen parameters is symptom disclosure; we excluded participants who did not respond to the EHS questions, potentially underestimating the true prevalence of erectile dysfunction. In typical clinical practice, physicians can elicit further information if patients omit EHS-related questions; however, such efforts may not be reflected in data-driven studies [128]. Moreover, evidence suggests that more than half of young men with erectile dysfunction do not disclose their symptoms, hinting at selection bias and underreporting. Finally, delayed ejaculation—commonly referred to as intravaginal ejaculatory dysfunction in Japan—may further complicate EHS interpretations, as it can arise from psychogenic factors (e.g., unrealistic sexual stimuli or inappropriate masturbation habits) [129]. Although this condition is widespread in Japan, it remains largely underexamined in other countries [130]. Overall, these factors underscore that the relatively low prevalence of erectile dysfunction observed here must be interpreted carefully, with an awareness of the sociocultural and methodological complexities that shape men’s sexual health outcomes.

### 4.4. Study Limitations and Future Directions

The results of this study advance our understanding of the complex interplay between endocrine function, trace elements, and symptom manifestation. Our approach—employing a broad range of biomarkers, comprehensive symptom evaluation, and data-driven clustering—supports the move toward more personalized health assessments. Such nuanced understanding may inform targeted interventions, guide efforts to maintain or restore optimal testosterone levels, improve men’s health, and potentially enhance societal productivity.

Nevertheless, this study has several limitations that warrant consideration. First, the single-center, cross-sectional design may limit the generalizability of our findings, particularly to populations beyond those seeking infertility treatment. Second, although we employed the AMS to gauge symptom severity, this instrument cannot fully exclude non-testosterone-related factors (e.g., major depressive disorder and anxiety), and it is not recommended as a standalone screening tool for assessing testosterone deficiency [131]. Third, we did not measure bioavailable testosterone or incorporate key proteins, such as sex hormone-binding globulin and albumin [132], which influence testosterone bioavailability and could refine our understanding. Fourth, we assumed linear relationships between trace elements and testosterone decline risk; however, trace elements may follow U- or J-shaped dose–response curves [133], necessitating more sophisticated modeling approaches that our sample size could not support [134]. Finally, we did not address stress-related ejaculatory dysfunction, which is influenced by factors such as Mg [135], due to the lack of relevant data (e.g., ejaculation-related metrics) in AMS or EHS assessments. Future studies should incorporate these considerations to achieve a more comprehensive understanding of testosterone-related health problems in young men.

## 5. Conclusions

Significant associations were observed between testosterone levels, symptom severity, and a range of serum trace elements in young men, suggesting that certain trace elements can mitigate or exacerbate the health consequences of declining testosterone levels. These findings may inform the development of personalized interventions and new standards for health assessments, ultimately improving male well-being and contributing to a more sustainable and productive society in the future.

## Figures and Tables

**Figure 1 nutrients-17-00867-f001:**
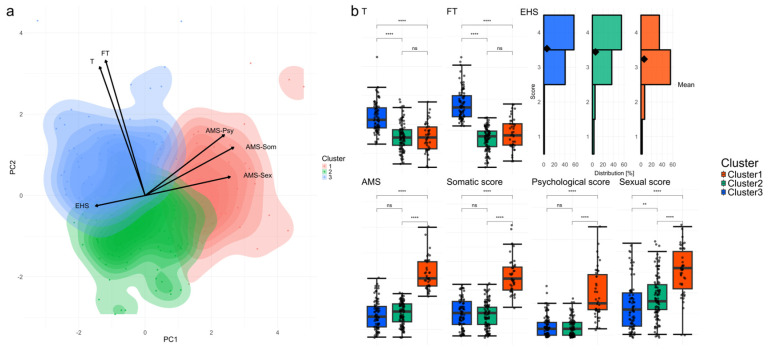
Clustering of participants by principal components and k-means, highlighting testosterone and symptom-related variables. (**a**) Principal component analysis performed using six explanatory variables and classification into three clusters using k-means values. T, total testosterone; FT, free testosterone; AMS; Aging Males’ Symptoms (scale); AMS-Som, somatic subdomain of AMS; AMS-Psy, psychological subdomain of AMS; AMS-Sex, sexual subdomain of AMS; EHS, Erection Hardness Score; PC1, first principal component (horizontal axis); PC2, second principal component (vertical axis). (**b**) Comparison of profiles for the clinical interpretation of the three clusters. The black diamonds in the EHS histogram represent the mean values. The Wilcoxon rank-sum test was used with Bonferroni correction for multiple comparisons. ns: not significant; ** *p* < 0.01; **** *p* < 0.0001.

**Figure 2 nutrients-17-00867-f002:**
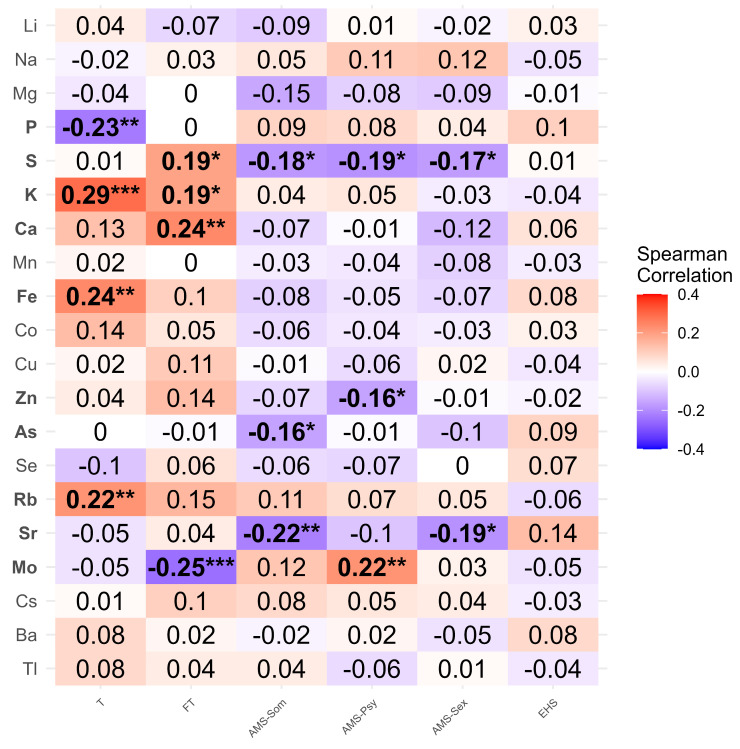
Correlation heatmap between serum trace elements and testosterone-related explanatory variables after adjusting for age. Heatmap of the correlation matrix between 20 serum trace element concentrations and 6 explanatory variables after adjusting for age. T, total testosterone; FT, free testosterone; AMS; Aging Males’ Symptoms (scale); AMS-Som, somatic subdomain of AMS; AMS-Psy, psychological subdomain of AMS; AMS-Sex, sexual subdomain of AMS; EHS, Erection Hardness Score. Values represent Spearman’s rank correlation coefficients (Spearman’s “rho”). *, *p* < 0.05; **, *p* < 0.01; ***, *p* < 0.001.

**Figure 3 nutrients-17-00867-f003:**
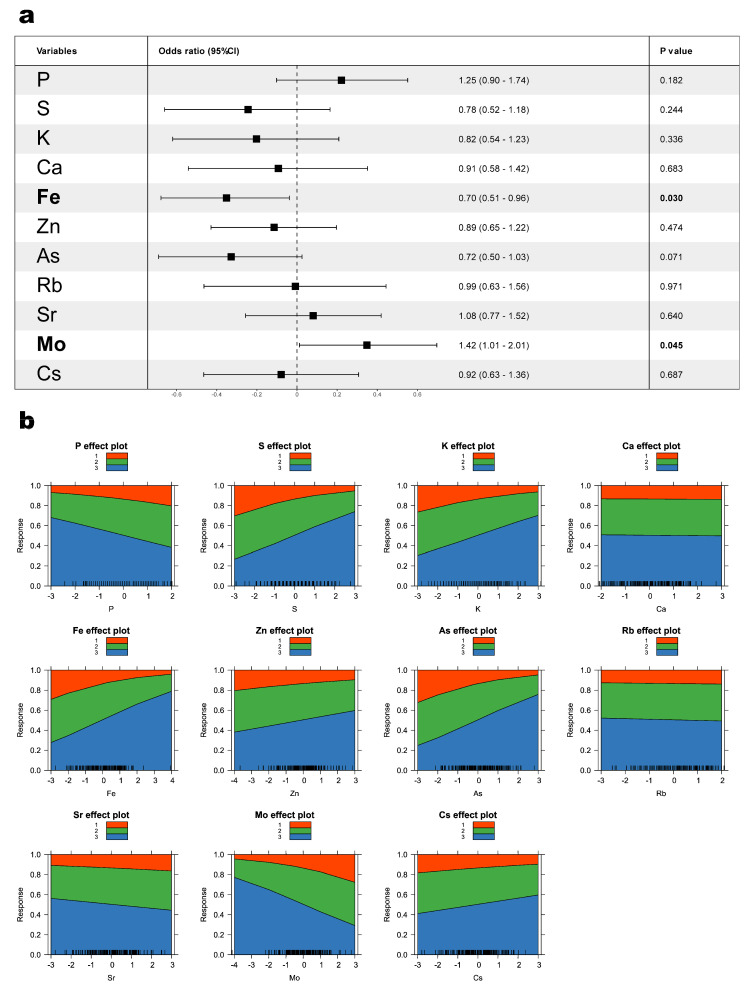
Ordinal logistic regression and logistic plots illustrating trace elements associated with cluster-defined testosterone levels. (**a**) This forest plot illustrates the results of an ordinal logistic regression, showing the impact of 11 trace elements, identified based on significant correlations, on the three clusters. Odds ratios and their 95% confidence intervals (CI), represented by black squares and bars, respectively, indicate the degree of contribution to Cluster 1. Higher odds ratio values on the right suggest a stronger association with Cluster 1, indicating lower testosterone levels and more severe symptoms. (**b**) Eleven logistic plots displaying the scaled serum concentrations of each trace element on the x-axis (standardized as Z-scores), with whiskers indicating the presence of cases. The y-axis represents the predicted cumulative probability of falling below the corresponding level for each trace element.

**Figure 4 nutrients-17-00867-f004:**
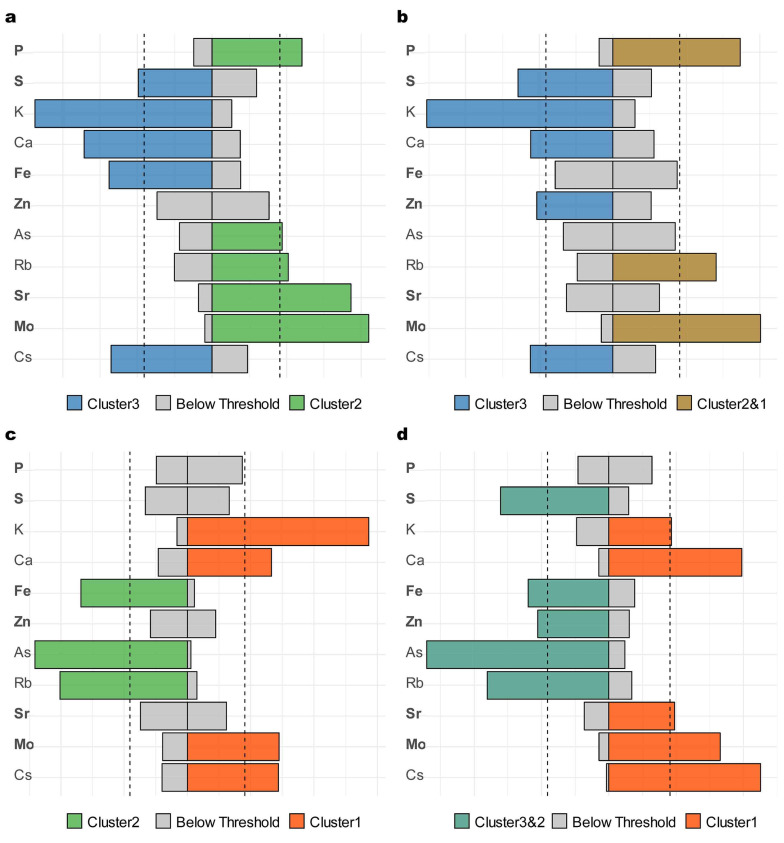
Weighted quantile sum regression identifying significant trace element contributors across testosterone-related cluster comparisons. Bar plots, resulting from bidirectional weighted quantile sum regression, show the weighted contributions of components in positive (left) and negative (right) directions for four comparative analyses: (**a**) high testosterone cluster (Cluster 3) versus intermediate cluster (Cluster 2); (**b**) high testosterone cluster (Cluster 3) versus combined intermediate and low clusters (Clusters 2 and 1); (**c**) intermediate cluster (Cluster 2) versus low testosterone cluster (Cluster 1); (**d**) combined high and intermediate clusters (Clusters 3 and 2) versus low testosterone cluster (Cluster 1). The vertical dashed lines represent the threshold τ (tau), which discriminates the components that represent the highest importance in the mixture effect. Components extending beyond τ are considered significant contributors to the respective associations.

**Table 1 nutrients-17-00867-t001:** Summary of the baseline characteristics of the study population.

	Median (IQR)	Range
Age (years)	37 (33–42)	22–75
Height (cm)	172.0 (168.0–176.0)	155.0–186.0
Weight (kg)	68 (62–77)	46–130
BMI (kg/m^2^)	23.0 (20.9–25.5)	17.6–44.5

IQR, interquartile range; BMI, body mass index.

**Table 2 nutrients-17-00867-t002:** Comparison of explanatory variables and SDS scores before and after age matching.

	Before Matching by Age				After Matching by Age			
	Cluster 1 (n = 43)	Cluster 2 (n = 98)	Cluster 3 (n = 84)	*p*-Value	Cluster 1 (n = 27)	Cluster 2 (n = 57)	Cluster 3 (n = 84)	*p*-Value
T (ng/mL)	3.87 (2.90–4.78)	3.87 (3.20–4.54)	5.40 (4.70–6.42)	**<0.001**	3.70 (3.01–4.48)	3.85 (3.22–4.43)	5.40 (4.70–6.42)	**<0.001**
FT (pg/mL)	9.2 (7.7–11.1)	9.1 (7.4–9.9)	13.7 (12.2–15.5)	**<0.001**	9.7 (7.7–11.0)	9.1 (7.6–9.6)	13.7 (12.2–15.5)	**<0.001**
AMS (points)	40 (37–47)	27 (23–30)	25 (21–29)	**<0.001**	40 (36–45)	25 (23–30)	25 (21–29)	**<0.001**
Somatic (points)	17 (15–19)	11 (9–12)	11 (9–13)	**<0.001**	17 (16–19)	11 (9–12)	11 (9–13)	**<0.001**
Psychological (points)	10 (9–15)	6 (5–7)	6 (5–7)	**<0.001**	10 (9–14)	6 (5–7)	6 (5–7)	**<0.001**
Sexual (points)	13 (11–15)	9 (8–11)	8 (6–10)	**<0.001**	12 (10–14)	9 (7–10)	8 (6–10)	**<0.001**
EHS (points)	3.23 (0.10)	3.44 (0.07)	3.54 (0.06)	**0.038**	3.41 (0.10)	3.47 (0.10)	3.54 (0.06)	0.4
Grade 1	2.3%	3.1%	1.2%		0%	3.5%	1.2%	
Grade 2	7.0%	5.1%	1.2%		0%	3.5%	1.2%	
Grade 3	56%	37%	40%		59%	35%	40%	
Grade 4	35%	55%	57%		41%	58%	57%	
SDS (points)	44 (40–48)	37 (31–40)	38 (33–41)	**<0.001**	44 (41–48)	38 (32–41)	38 (33–41)	**<0.001**
Affective (points)	16 (15–18)	12 (10–14)	13 (11–14)	**<0.001**	16 (15–18)	12 (11–14)	13 (11–14)	**<0.001**
Cognitive Score (points)	18 (15–19)	15 (13–17)	16 (13–18)	**<0.001**	18 (15–19)	16 (13–17)	16 (13–18)	**0.017**
Somatic Score (points)	11 (9–12)	9 (7–11)	9 (8–10)	**<0.001**	11 (9–12)	9 (7–11)	9 (8–10)	**<0.001**

T, total testosterone; FT free testosterone; AMS, Aging Males’ Symptoms (scale); EHS, Erection Hardness Score; SDS, Self-Rating Depression Scale. The EHS is presented as the mean (standard error), and the other outcomes are presented as the median (interquartile range). The *p*-values were based on the Kruskal–Wallis test. The SDS had 11 missing values before and 10 after age matching and was therefore excluded from the analysis. Bold *p*-values indicate statistical significance (*p* < 0.05).

**Table 3 nutrients-17-00867-t003:** Comparison of semen parameters and secondary outcomes before and after age matching.

	Before Matching by Age				After Matching by Age			
	Cluster 1 (n = 43)	Cluster 2 (n = 98)	Cluster 3 (n = 84)	*p*-Value	Cluster 1 (n = 27)	Cluster 2 (n = 57)	Cluster 3 (n = 84)	*p*-Value
Age (years)	35 (33–41)	39 (34–44)	36 (32–39)	**0.007**	34 (32–39)	37 (32–40)	36 (32–39)	0.5
Height (cm)	171.0 (168.0–174.0)	170.8 (168.0–174.0)	174.0 (170.0–177.0)	**0.010**	170.0 (168.0–173.0)	171.0 (168.0–174.0)	174.0 (170.0–177.0)	**0.013**
Weight (kg)	70 (62–80)	67 (60–76)	67 (63–75)	0.7	68 (59–81)	70 (60–79)	67 (63–75)	>0.9
BMI (kg/m^2^)	23.5 (21.3–27.2)	22.9 (20.7–25.9)	23.0 (21.1–24.6)	0.4	23.3 (20.9–26.7)	23.0 (20.8–26.6)	23.0 (21.1–24.6)	0.8
SV (mL)	3.10 (2.05–3.95)	3.50 (2.60–4.30)	3.60 (2.90–4.93)	0.085	3.20 (2.70–4.10)	3.70 (2.60–4.70)	3.60 (2.90–4.93)	0.4
SC (×10^6^ mL)	36 (7–82)	40 (14–102)	41 (8–89)	0.5	25 (4–58)	42 (14–105)	41 (8–89)	0.2
SM (%)	52 (35–67)	51 (31–68)	55 (38–66)	0.8	53 (35–67)	54 (30–69)	55 (38–66)	0.9
Left TV (mL)	22 (16–24)	20 (16–22)	20 (18–22)	0.5	21 (16–23)	20 (16–22)	20 (18–22)	>0.9
Right TV (mL)	22 (18–24)	20 (18–22)	20 (18–24)	0.7	21 (18–23)	20 (18–24)	20 (18–24)	0.9
LH (mIU/mL)	2.66 (1.84–3.53)	2.58 (1.79–3.69)	2.70 (2.15–3.60)	0.4	2.71 (1.84–3.47)	2.47 (1.67–3.50)	2.70 (2.15–3.60)	0.3
FSH (mIU/mL)	4.5 (3.0–6.1)	4.3 (2.8–6.7)	4.3 (3.2–6.0)	>0.9	4.9 (2.9–6.1)	4.0 (2.7–5.8)	4.3 (3.2–6.0)	0.4
PRL (ng/mL)	7.6 (5.5–10.0)	8.0 (6.0–10.8)	8.3 (6.3–10.1)	0.5	7.9 (5.8–11.2)	8.2 (6.5–10.5)	8.3 (6.3–10.1)	>0.9
E2 (pg/mL)	18 (13–26)	17 (12–22)	20 (14–24)	0.3	20 (16–27)	18 (12–22)	20 (14–24)	0.6

BMI, body mass index; SV, semen volume; SC, sperm concentration, SM, sperm motility; TV, testicular volume; LH, luteinizing hormone; FSH, follicle-stimulating hormone; PRL, prolactin; E2; estradiol. Outcomes are presented as median (interquartile range). The *p*-values were based on the Kruskal–Wallis test. Bold *p*-values indicate statistical significance (*p* < 0.05)

**Table 4 nutrients-17-00867-t004:** Comparison of serum trace elements among the clusters after age-matching.

	Cluster 1 (n = 27)	Cluster 2 (n = 57)	Cluster 3 (n = 84)	*p*-Value
Li (μg/L)	0.46 (0.38–0.57)	0.48 (0.42–0.62)	0.50 (0.35–0.67)	0.6
Na (mg/L)	3180 (3140–3210)	3160 (3110–3200)	3160 (3085–3200)	0.12
Mg (mg/L)	20.00 (18.80–20.55)	19.90 (19.00–20.60)	19.95 (18.90–20.78)	0.9
P (mg/L)	119 (114–131)	116 (112–127)	118 (108–129)	0.6
**S (mg/L)**	1130 (1105–1150)	1130 (1090–1170)	1150 (1118–1190)	**0.029**
**K (mg/L)**	163 (161–168)	159 (156–168)	167 (158–174)	**0.014**
**Ca (mg/L)**	93.6 (91.5–95.7)	92.1 (89.9–94.6)	94.6 (92.0–96.6)	**0.009**
Mn (μg/L)	0.47 (0.39–0.55)	0.47 (0.41–0.54)	0.49 (0.41–0.59)	0.4
Fe (μg/L)	912 (777–1245)	1030 (811–1220)	1075 (895–1368)	0.10
Co (μg/L)	0.086 (0.075–0.099)	0.082 (0.074–0.098)	0.089 (0.079–0.104)	0.3
Cu (μg/L)	718 (657–790)	741 (682–794)	757 (676–828)	0.4
Zn (μg/L)	749 (702–844)	784 (694–864)	829 (731–895)	0.080
**As (μg/L)**	0.85 (0.67–1.33)	1.35 (0.94–1.97)	1.09 (0.75–2.41)	**0.019**
Se (μg/L)	146 (138–156)	141 (135–148)	143 (135–154)	0.2
**Rb (μg/L)**	175 (152–185)	161 (151–183)	176 (159–197)	**0.049**
Sr (μg/L)	28 (21–33)	26 (23–32)	27 (22–33)	0.8
Mo (μg/L)	1.25 (0.82–1.80)	1.22 (0.92–1.76)	1.01 (0.86–1.42)	0.12
**Cs (μg/L)**	0.64 (0.58–0.74)	0.56 (0.50–0.72)	0.66 (0.56–0.76)	**0.035**
Ba (μg/L)	0.45 (0.35–0.68)	0.36 (0.29–0.50)	0.42 (0.31–0.60)	0.060
Tl (μg/L)	0.036 (0.030–0.044)	0.035 (0.027–0.044)	0.037 (0.032–0.044)	0.7

The variables are presented as medians (interquartile ranges). The *p*-values were based on the Kruskal–Wallis test. Bold values indicate statistically significant differences (*p* < 0.05) across clusters.

## Data Availability

The raw data supporting the conclusions of this article will be made available by the authors on request.

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
