# Peer review of "Distinct Clusters of Testosterone Levels, Symptoms, and Serum Trace Elements in Young Men: A Cross-Sectional Analysis"

_nutrients, 2025, doi:10.3390/nu17050867_

Round 1

Reviewer 1 Report

Comments and Suggestions for Authors

Review of the manuscript entitled: Distinct Clusters of Testosterone Levels, Symptoms, and Serum Trace Elements in Young Men: A Cross-Sectional Analysis (ID: nutrients-3470529). Some of my suggestions for the manuscript:

  1. In the abstract, the authors should clearly state the purpose of the study.
  2. “Associations were analyzed with Spearman’s rank correlation, ordinal logistic regression, and Weighted Quantile Sum regression, adjusting for age” is this sentence needed in the abstract?
  3. In the introduction, the authors should clearly state the purpose of the study. 
  4. “To achieve this, we applied unsupervised clustering [27] to classify men based on serum total and free testosterone levels and symptom severity without relying on arbitrary cut-off values. We then examined the differences in serum trace element concentrations among these clusters to identify potential associations” is this sentence needed in the introduction? In my opinion, in the introduction, the Authors should discuss the current knowledge on the effect of trace elements on testosterone levels and then indicate what gaps in knowledge their study fills. I ask the Authors to supplement the introduction. 
  5. The Authors have chosen a non-standard way of presenting the study methodology in the supplementary materials to the manuscript. I must admit that this way of presenting the methodology is tiring for me as a reader/reviewer, because I have to read two articles to find out what the study methodology was. I believe that the study methodology should be presented in the main text in a clear, concise and reader-friendly way according to the STROBE scheme. The supplementary materials should include information that goes beyond the main text and may be important from the research point of view. Of course, what is in the supplementary materials regarding the study methodology should be presented concisely in the main text.
  6. The authors use an incorrect way of writing the cited literature, line: 101-102: "[30,31]<Chang, 2018, PMID: 30363972; Baudry, 2019, PMID: 31786641."
  7. I emphasize once again that the way of presenting the research methodology is complicated and in two files, that I as a reader do not know where to look for information. This requires correction and presentation in one main text. The authors should remember that their article is addressed to professionals. There is no need to describe everything in such detail, also in the form of an appendix. I suggest the authors see how others did it in already published articles.
  8. "Trace element levels were quantified using methods previously described in the literature [45-47]" what method was that? I have to search the literature to find out if the authors indicate at the end of the sentence which methods were used? It is explained later in the paragraph what the method was, but such a record confuses the reader.
  9. The results are presented correctly.
  10. The discussion is also sufficient in my opinion.
  11. In summary, the results presented in the manuscript are valuable, but the authors need to improve the way the research methodology is presented.

Author Response

Comments 1: In the abstract, the authors should clearly state the purpose of the study.

Response 1: Thank you for the careful review and pertinent comments. The purpose of this study is clearly stated in this sentence: “The study aim was to examine how serum trace elements modulate the relationship between testosterone levels and symptom severity.”

Comments 2: “Associations were analyzed with Spearman’s rank correlation, ordinal logistic regression, and Weighted Quantile Sum regression, adjusting for age” is this sentence needed in the abstract?

Response 2: We have streamlined the Methods section by mentioning “unsupervised clustering” and stating that we “compared” trace elements among clusters. This reduces excessive detail in the abstract while retaining essential methodological information.

Comments 3: In the introduction, the authors should clearly state the purpose of the study.

Response 3: We agree with the reviewer that the Introduction would benefit from a clearer statement of the study purpose. We have added a clear objective toward the end, specifying that the study aimed to examine how trace elements influence the relationship between testosterone levels and symptom severity in young men.

Comments 4: “To achieve this, we applied unsupervised clustering [27] to classify men based on serum total and free testosterone levels and symptom severity without relying on arbitrary cut-off values. We then examined the differences in serum trace element concentrations among these clusters to identify potential associations” is this sentence needed in the introduction? In my opinion, in the introduction, the Authors should discuss the current knowledge on the effect of trace elements on testosterone levels and then indicate what gaps in knowledge their study fills. I ask the Authors to supplement the introduction.

Response 4: Thank you for your recommendation to focus the Introduction more on the existing literature regarding trace elements and testosterone. To clarify the knowledge gap, in the revised Introduction, we have:

  1. Expanded our discussion of trace elements (e.g., iron, zinc, copper, manganese) and their roles in testosterone regulation and mental health, highlighting existing evidence on how these elements may either exacerbate or alleviate symptoms related to low testosterone.
  2. Clearly stated the knowledge gap, specifically that few comprehensive, multi-element studies have explored how trace elements collectively relate to testosterone levels and health outcomes in younger or subfertile men.
  3. Moved specific methodological details about unsupervised clustering to the Methods section, with only a brief mention in the Introduction. This allows the Introduction to focus on synthesizing current knowledge, identifying gaps, and outlining the study aims.

We believe that these revisions clarify how our study addresses the gap in understanding the interplay between testosterone levels, symptom severity, and a panel of 20 serum trace elements, particularly in younger men likely exposed to modern-day stressors.

Comments 5: The Authors have chosen a non-standard way of presenting the study methodology in the supplementary materials to the manuscript. I must admit that this way of presenting the methodology is tiring for me as a reader/reviewer, because I have to read two articles to find out what the study methodology was. I believe that the study methodology should be presented in the main text in a clear, concise and reader-friendly way according to the STROBE scheme. The supplementary materials should include information that goes beyond the main text and may be important from the research point of view. Of course, what is in the supplementary materials regarding the study methodology should be presented concisely in the main text.

Response 5: Thank you for your constructive comments regarding the presentation of our methodology. We apologize for any inconvenience caused. In response to your feedback, we have removed the supplementary materials and incorporated all essential methodological details into the main text.

Comments 6: The authors use an incorrect way of writing the cited literature, line: 101-102: "[30,31]<Chang, 2018, PMID: 30363972; Baudry, 2019, PMID: 31786641."

Response 6: Thank you for bringing this to our attention. These bracketed notes were added for our reference and, unfortunately, were not removed before submission. We have now deleted them and ensured that the citation style is correctly presented in the revised manuscript.

Comments 7: I emphasize once again that the way of presenting the research methodology is complicated and in two files, that I as a reader do not know where to look for information. This requires correction and presentation in one main text. The authors should remember that their article is addressed to professionals. There is no need to describe everything in such detail, also in the form of an appendix. I suggest the authors see how others did it in already published articles.

Response 7: Thank you for highlighting this important issue. We greatly appreciate your careful attention to the presentation of our methodology. As you rightly noted, an overly detailed description of methods can be redundant for professionals. In response to your comments, we have kept the essential methodological details in the main text and removed the supplementary materials with additional information. Since this journal does not impose a strict word limit, we have also integrated the discussion content from the supplementary materials into the main text. We believe that the resulting detailed literature review and discussion will be highly beneficial to our readers.

Comments 8: "Trace element levels were quantified using methods previously described in the literature [45-47]" what method was that? I have to search the literature to find out if the authors indicate at the end of the sentence which methods were used? It is explained later in the paragraph what the method was, but such a record confuses the reader.

Response 8: Thank you for pointing this out. We understand that referencing our previously published methods without explicitly stating “ICP-MS” upfront may have caused confusion. We have revised the Methods section to clearly specify at the outset that inductively coupled plasma mass spectrometry (ICP-MS) was used to quantify trace elements. We have also retained references to our earlier work [49–51] for those interested in the detailed calibration procedures and instrument settings. This ensures that readers can immediately identify the technique without needing to consult external sources, while also avoiding undue repetition (and the risk of self-plagiarism) and still providing sufficient information for reproducibility.

Comments 9: The results are presented correctly.

Response 9: Thank you for carefully reviewing our results. We appreciate your positive feedback.

Comments 10: The discussion is also sufficient in my opinion.

Response 10: Thank you for your positive feedback. As mentioned earlier, we have integrated the supplementary file containing an extensive literature review into the main text to provide better support and guidance to our readers.

Comments 11: In summary, the results presented in the manuscript are valuable, but the authors need to improve the way the research methodology is presented.

Response 11: We would like to note that we have made every effort from the outset to present our methodology in accordance with the STROBE guidelines, and as such, no substantial revisions were made to this section. We appreciate your constructive feedback and look forward to your continued support.

Reviewer 2 Report

Comments and Suggestions for Authors

This paper presents the results of a study considering the role of trace elements as factors that mitigate or exacerbate the health consequences of reduced testosterone levels in young men in Japan. The study pointed out the important role of trace elements such as P, Sr, Mo, Fe, S and Zn in this regard. This is a very great achievement for the Authors, since trace elements are rarely considered in various types of clinical studies, and their importance for human health is difficult to overestimate.

The study was well designed and performed, and the results were presented in the form of easy-to-read Figures and Tables (also in Supplementary materials). A well-organized discussion of the results of the study served to develop definite conclusions. Also valuable to the discussion are the several limitations that warrant consideration, as indicated by the Authors. Before publishing the paper, however, the authors should address several suggestions below.

Lines 201-203, this section of text should be removed from the manuscript.

Lines 324-333, The authors write about 11 trace elements, while Figure 3b shows logistic plots for only 10 trace elements (no Cs).

Several of the cited articles are written in a language other than English (43, 46, 68), so their content will not be understood by most Nutrients journal readers. Would it be possible to replace them with English-language publications?

Author Response

Comments 1: This paper presents the results of a study considering the role of trace elements as factors that mitigate or exacerbate the health consequences of reduced testosterone levels in young men in Japan. The study pointed out the important role of trace elements such as P, Sr, Mo, Fe, S and Zn in this regard. This is a very great achievement for the Authors, since trace elements are rarely considered in various types of clinical studies, and their importance for human health is difficult to overestimate.

The study was well designed and performed, and the results were presented in the form of easy-to-read Figures and Tables (also in Supplementary materials). A well-organized discussion of the results of the study served to develop definite conclusions. Also valuable to the discussion are the several limitations that warrant consideration, as indicated by the Authors. Before publishing the paper, however, the authors should address several suggestions below.

Response 1: Thank you for the careful review and pertinent comments. We are grateful for your fair and positive evaluation of our research achievements. Below, you will find our detailed responses to your comments.

Comment 2: Lines 201-203, this section of text should be removed from the manuscript.

Response 2: Thank you for your valuable comment. It appears that we inadvertently left the template text provided by the publisher in our manuscript. We have now removed lines 201–203 as suggested.

Comment 3: Lines 324-333, The authors write about 11 trace elements, while Figure 3b shows logistic plots for only 10 trace elements (no Cs).

Response 3: Thank you for your careful observation. It appears that Cs was inadvertently omitted from the coding for this analysis. We have recoded the analysis and replaced Figure 3b with the corrected version that now includes Cs. Additionally, due to the inherent characteristics of R (such as seed values and package version variations), there are slight numerical differences compared to our previous outputs. We have re-created the other figures accordingly to maintain consistency. We believe that these minor differences will not affect the overall conclusions, and we appreciate your understanding.

Comment 4: Several of the cited articles are written in a language other than English (43, 46, 68), so their content will not be understood by most Nutrients journal readers. Would it be possible to replace them with English-language publications?

Response 4: Thank you for your astute observation and for sharing your concern. We understand why you might question the inclusion of articles written in languages other than English. However, we believe it is important to distinguish between the relatively limited demand for non-English sources among most readers and the practice of excluding such references from the scientific record. These references have been integral to our work, and we have carefully adapted relevant information from Japanese to English, ensuring that there is no appearance of plagiarism. Nevertheless, to avoid any doubts about originality, we believe it is necessary to properly cite these sources, regardless of the language in which they were published.

Furthermore, cross-lingual citation is widely recognized by scholars as an essential way to prevent the "siloing" of scientific discoveries across linguistic barriers (see https://doi.org/10.1007/s00799-021-00312-z). With the rise of online journals and advancements in machine translation, non-Japanese-speaking researchers now have more opportunities to access and benefit from Japanese-language articles (see https://doi.org/10.1093/biosci/biac062). Promoting cross-lingual knowledge sharing in this way has also been a key focus of our team’s previous initiatives (see https://doi.org/10.31662/jmaj.2024-0140). We hope this explanation helps clarify our decision to retain these references in our manuscript, and we appreciate your understanding.